



# Simulating stress-dependent fluid flow in a fractured core sample using real-time X-ray CT data

Tobias Kling[1], Da Huo[2], Jens-Oliver Schwarz[3,4], Frieder Enzmann[3], Sally Benson[2], Philipp Blum[1]

[1]Institute for Applied Geosciences (AGW), Karlsruhe Institute of Technology (KIT), Karlsruhe, 76131, Germany
[2]School of Earth Sciences, Stanford University, Stanford, CA, 94305-2210, United States of America (USA)
[3]Institute for Geosciences, Johannes Gutenberg University of Mainz, Mainz, 55128, Germany
[4]Math2Market GmbH, Kaiserslautern, 67657, Germany

*Correspondence to*: Tobias Kling (tobias.kling@kit.edu)

**Abstract.** The objective of the current study is to investigate and validate stress-dependent single fluid flow in a fractured
core sample using in situ X-ray computed tomography (CT) scans and a finite-volume method solving the Navier-Stokes-Brinkman equations. The permeability of the fractured sandstone sample was measured stepwise during a loading-unloading cycle (0.7 MPa to 22.1 MPa and back) to validate the numerical results. Simultaneously, the pressurized core sample was imaged with a medical X-ray CT scanner with a voxel dimension of $0.5 \times 0.5 \times 1.0$ mm³. Fracture geometries were obtained by CT images based on the Missing Attenuation (MA) approach. Simulation results revealed both, qualitative plausibility
and a quantitative approximation of the experimentally derived permeabilities. The qualitative results indicate flow channeling along several preferential flow paths with less pronounced tortuosity. Significant changes in permeability can be assigned to temporal and permanent changes within the fracture due to applied stresses. The applied fluid flow simulations also incorporate potential fracture-matrix interaction and permeability anisotropy within the matrix caused by high-porosity layers. The deviations of the quantitative results appear to be mainly caused by the low resolution affecting the accurate
capturing of sub-grid scale features and the reproduction of the actual connectivity. Furthermore, the threshold value $CT_{mat}$ (1862.6 HU) depicting the matrix material represents the most sensitive input parameter of the simulations. Small variations of $CT_{mat}$ ($\pm17.7$ HU in this study) can cause enormous changes in simulated permeability by up to a factor of $2.6 \pm 0.1$ and, thus, has to be defined with caution. Finally, our results are also compared with other studies showing similar results. Based on these observations various recommendations to improve CT image quality, model quality, aperture calibration and
validation of qualitative fluid flow are provided.

Keywords: Fracture flow, Confining pressure, Medical X-ray CT, Numerical analysis, Navier-Stokes-Brinkman



## 1 Introduction

Naturally and artificially induced hydromechanical coupling is essential for the understanding of many geologic processes within the Earth's crust and for the successful realization of a wide range of geoscientific applications. A more detailed overview about these applications such as geothermal energy generation, nuclear waste disposal or hydrocarbon production was presented by Rutqvist and Stephansson (2003). Since there are different phenomena of direct and indirect hydromechanical coupling in geosciences, this study only deals with the direct solid-to-fluid coupling, which is defined as the stress induced changes in fluid dynamics (Wang, 2000).

In the past, the stress-dependency of fracture permeability and its hysteretic behavior due to stepwise loading and subsequent unloading was investigated by various authors (e.g. Gangi, 1978; Kranz et al., 1979; Snow, 1965). Various empirical models were developed that approximate stress-dependent fracture permeability by adding roughness-based variables to the common cubic law approach (Bernabe, 1986; Gale, 1982; Gangi, 1978; Huo and Benson, 2015; Swan, 1983; Walsh, 1981; Witherspoon et al., 1980; Zimmerman et al., 1992). However, flow in rough fractures is governed by several additional morphology-related features such as the grade of connectivity, variations in tortuosity, flow separation and stagnant zones (Konzuk and Kueper, 2002; Tsang, 1984) that are hard to empirically quantify.

Hence, numerical simulations that allow the implementation of much more sophisticated physical solution approaches and boundary conditions represent a powerful tool in modern geosciences. Accordingly, there is an immense number of Computational Fluid Dynamics (CFD) studies concerning different topics such as single-phase and multi-phase flow (Sahimi, 2011), natural (Crandall et al., 2010) and artificial fractures (Brush and Thomson, 2003) or fractured porous media (Landry and Karpyn, 2012). The same applies to CFD methods, that are most frequently represented by finite-difference (FD), finite-element (FE), lattice gas and lattice Boltzmann (LB) methods (Madadi and Sahimi, 2003). In particular, the lattice methods are suggested by Madadi and Sahimi (2003) as the ideal computational base for arbitrary geometries. In addition to that, the finite-volume (FV) method has been increasingly and successfully used to simulate fracture flow in the last few years (Al-Yaarubi et al., 2005; Brush and Thomson, 2003; Crandall et al., 2010; Huber et al., 2012; Schwarz and Enzmann, 2013). However, in practice solving three-dimensional mathematical models such as the Navier-Stokes equation with these methods can be computationally elaborate so that many simulations prefer simplified flow models such as the "local cubic law" approach (Brush and Thomson, 2003; Konzuk and Kueper, 2004; Koyama et al., 2008; Oron and Berkowitz, 1998; Zimmerman and Yeo, 2013).

Various studies concerning numerical simulations of stress-dependent fracture flow in core-scale dimensions have been performed until now. Table 1 provides a chronological list of previously performed studies. All these simulations focus on dynamic changes of fracture permeability due to loading conditions. However, they do not address residual changes as a





result of mechanical deformation within the fracture, which can be addressed by subsequent unloading. Except the study by Watanabe et al. (2008), who highlight qualitative insights into the flow structures, all other studies in Table 1 quantify and intend to validate simulated fracture flow. In a subsequent study which is not listed in Table 1, they applied a similar (non-quantitative) procedure for other fracture types (Watanabe et al., 2009). By contrast, quantification of fracture flow was

performed by Indraratna et al. (2015), who performed CFD simulations based on initial laser scanning measurements, where the flow model is coupled with a deformation criteria simulating fracture closing. Similar, but more simplified deformation procedures were performed for statistically derived aperture distributions (Pyrak-Nolte and Morris, 2000), profilometer measurements (Kim et al., 2003), laser-scan data (Nemoto et al., 2009; Watanabe et al., 2008, 2009) and a conceptual 2D model (Liu et al., 2010). Furthermore, Dicman et al. (2004) described changes in aperture distribution by combining

laboratory data and stochastic approaches. Considering this study and further information provided in Table 1, most of the used fracture geometries can be assumed to be not directly correlated to the actual fracture geometries under loading conditions used for validation. For this reason, more recent studies introduced X-ray computed tomography (CT) that enables the scanning of in situ conditions during the experiment within the sample and creates "real-time" images of the fracture (Watanabe et al., 2011; Watanabe et al., 2013).

In the past, in situ imaging of experiments based on CT technologies was applied to a wide range of studies including geomechanical behavior of fractures (Re and Scavia, 1999), evaluations of triaxial tests (Feng et al., 2004; Ge et al., 2001; Lenoir et al., 2007; Ren and Ge, 2004; Vinegar et al., 1991; Zhou et al., 2008), shear tests (Tatone and Grasselli, 2015), sand production experiments (Santos et al., 2010) as well as various single-/multi-phase core-flooding experiments in fractured

and unfractured rocks (Huo and Benson, 2015; Krevor et al., 2012; Oh et al., 2013; Perrin and Benson, 2010; Pini and Benson, 2013; Pini et al., 2012; Rangel-German et al., 2006; Schembre and Kovscek, 2003; Shi et al., 2009; Watanabe et al., 2011). CT technology generally represents a three-dimensional (3D), non-destructive method to image density contrasts in high resolution and, therefore, is well suited to reproduce dynamic processes in situ and in real-time. In this study, the term "real-time" refers to a non-continuous but step-wise imaging of a sample being subjected to a dynamic process.

Comparing other appliances such as industrial, micro- or synchrotron-based CTs, medical CT scanners admittedly operate with the lowest resolutions. However, they generally have the advantage of lower scanning times and are more flexible concerning with sample/equipment weights and sizes compared (Watanabe et al., 2011). Currently, medical CT scanners represent the most economical option to conduct such experiments and also provide, particularly in regard to their

importance in medicine, a good accessibility.

The central issue concerning CT measurements and fractured rocks is given by the aperture calibration based on measured density contrasts. Considering a transect perpendicular to the fracture plane, there are generally four methods to calibrate apertures caused by the fracture-related density anomaly (Ketcham and Carlson, 2001; Ketcham et al., 2010): (1) The Peak



Height (PH) method, based on the gap between the idealized matrix density and the negative peak of the anomaly, which was successfully applied by Watanabe et al. (2011), but typically is more applicable for homogeneous materials (Ketcham et al., 2010). The PH method requires a careful calibration. (2) The Missing Attenuation (MA) method, which integrates the entire anomaly area, which was refined for smaller apertures, and can be used for heterogeneous rocks and, theoretically, do not require calibration (Huo et al., (2016)). (3) The full-width-half-maximum (FWHM) method using the midpoint between matrix and peak value; however, it is only applicable to larger apertures (Ketcham, 2010) and less accurate than the MA method (Van Geet and Swennen, 2001). (4) The inverse point-spread function (IPSF) method representing a hybrid between PH and MA methods and iteratively reconstructs fractures by using a PSF accounting for the blurring of the CT image. Indeed, IPSF method is well suitable to calculate apertures and also is applicable for heterogeneous materials. However, this method is computationally expensive due to the complex numerical deconvolution (Ketcham et al., 2010). Indeed, Watanabe et al. (2011) introduced a useful method to simulate stress-dependent fluid flow based on CT images and in fracture-scale before. However, their method revealed several limitations: (A) Simulations are very sensitive to image noise due to the core holder, (B) the aperture calibration that is based on the PH method needs a careful and very time-consuming calibration, and (C) the applied CFD simulation relies on a simplified fluid dynamics represented by the "local cubic law" approach. Thus, in order to be also applicable for subsequent multiphase and/or thermo-hydraulic-chemical-mechanical (THCM) processes, fracture flow simulations should combine both, most realistic (unimpeded) fracture geometries and physically sophisticated fluid dynamics, ideally without any great effort.

The objective of this study is therefore to propose a novel and practical approach based on real-time X-ray CT data to numerically validate and reproduce stress-dependent single-phase fluid flow in a fractured sandstone sample. This simulation approach consists of (1) an alternative strategy to improve image quality, (2) a novel aperture calibration approach for 3D simulations based on the MA method and (3) a sophisticated simulation method that accounts for fracture and matrix flows by solving the Navier-Stokes-Brinkman equation. To demonstrate the effectiveness of this method and to indicate also the transferability to real reservoir rocks such as an enhanced geothermal system (EGS), a fractured low permeable sample is used. In the present study, the fluid flow is simulated under loading and unloading conditions to prove the extent to which it is possible to reproduce flow characteristics and residual fracture changes. Imaging is performed by applying a medical CT simultaneously to corresponding core-flooding experiments allowing the validation of the simulated results. Furthermore, current simulations are compared to corresponding studies (Watanabe et al., 2011; Watanabe et al., 2013) providing various insightful recommendations for the successful implementations of future CT-based CFD simulations.



## 2 Material and methods

### 2.1 Sample

In this study a Zenifim Sandstone is used, which was obtained from the former (unproductive) hydrocarbon prospection well Ramon-1 located in the central Negev area in Southern Israel (Figure 1). The rock belongs to the sea marginal deposits of the

arkosic Zenifim Formation (Precambrian age) and originates from 1770 m below surface. Additional information on geology and stratigraphy can be obtained by Weissbrod and Sneh (2002). According to Huo and Benson (2015), who studied an analogue sample in their study, the sample is sedimentologically classified as an immature feldspatic greywacke. The sample is composed of poorly sorted and rounded grains with a mean grain size of 0.2 mm. The matrix is cemented by quartz resulting in low mean permeability of $5.92 \times 10^{-19}$ m² and porosity of 2.5 to 3.9 % with a bulk density of 2,490 kg m$^{-3}$

(Mercury Injection Capillary Pressure, MICP, analysis). However, microscopic analyses (Huo and Benson, 2015) and visible fine laminations with a thickness of a few millimeters, that dip towards the core axis, indicate a significant porosity heterogeneity within the core.

The utilized rock sample is a continuously cylindrical core sample with a length of 6.7 cm and a diameter of 5.0 cm (Table 1,

Figure 1). Before the coupled experiments, the sample was bisected along the core axis by generating a saw-cut fracture intersecting the previously mentioned laminations resulting in a discontinuity, which can be described as a (mated) smooth-walled and relatively tight fracture. It should be noted that abrasion processes and local grain disruptions due to the sawing process result most likely in some minor artefacts along the fracture surface.

### 2.2 Experiment

An aluminum core-holder was used for the core-flooding experiment (Figure 2). Contemporaneously, the sample was imaged using medical X-ray CT. This experimental setup is based on the a porosity heterogeneity study by Perrin and Benson (2010) and was modified and successfully adapted for core-flooding experiments in porous (Pini et al., 2012) and fractured rocks (Huo and Benson, 2015). The fractured core is positioned in an aluminum core-holder with alternating interlayers of Viton rubber, heat-shrinkable Teflon and nickel foil that can also be seen on the CT image (Figure 3). For

further technical details please refer to Huo and Benson (2015), who used the same experimental setup.

Permeability measurements of the core sample are based on the steady-state method. After saturating the encapsulated sample, three different flow rates (12, 16 and 20 ml/min) are successively applied. Within the scope of this study, the applied flow rates and corresponding pressure drops typically reveal a linear relationship indicating laminar fluid flow.

Consequently, permeability can be determined by using Darcy's law (Huo and Benson, 2015).



According to this, permeabilities are determined under stepwise changes of effective stress (σ′). Here, σ′ is defined as the difference between applied confining pressure and pore pressure. In order to characterize stress-dependency of the fractured sample comprehensively, changes in σ′ represent a full loading-unloading cycle. Following this, σ′ is increased stepwise from 0.7 MPa (2.1, 3.5, 5.5, 11.0 MPa) to 22.1 MPa under loading conditions and, subsequently, are decreased by applying equivalent stress intervals.

Contemporaneously, the core-holder is positioned in a medical X-ray CT scanner (General Electric Hi-Speed CT/I X-ray computed tomography) to reveal real-time images of the sample for every stress stage. Scans were performed at an energy level of 120 keV, a tube current of 200 mA and a display field of view of 25 cm. We use CT scans to obtain an x-y resolution in the plane of $0.5 \times 0.5$ mm² and a slice thickness of 1 mm resulting in corresponding voxel dimensions. As a result of the CT scans each voxel is assigned to a specific CT number in Hounsfield unit (HU). Furthermore, at each stress stage, multiple (five) scans are conducted and averaged afterwards, representing a practical method to reduce the random noise of CT scans (Huo et al., (2016); Pini et al., 2012).

### 2.3 Image processing

The geometry of the model is based on the averaged, multiple CT scan revealing a cylindrical set with a total dimension of $256 \times 256 \times 68$ voxels that still contains the multilayer construction of the core-holder as well as the filter plates of the fluid in- and outlet. Full processing is performed by using a customized MATLAB code. After reading the five data sets, every CT scan is resampled to an isotropic voxel size (one voxel of $0.5 \times 0.5 \times 1.0$ mm³ to four voxels of $0.25 \times 0.25 \times 0.25$ mm³). Subsequently, the five scans are averaged to a single image (Figure 3). In a further step, geometric information stored by single CT numbers are transformed to according geometric (local apertures) and hydraulic (local permeabilities) properties being essential for the flow simulations. The corresponding calibration approach is explained fully in the next paragraph. Finally, the core-holder and filter plates are numerically cropped based on known core dimensions and obvious density contrasts resulting in a final sample dimension of $194 \times 194 \times 258$ voxels. Note that marginal voxels of the sample are directly affected by the adjacent core-holder so that the processed sample is slightly smaller than expected ($200 \times 200 \times 268$ voxels).

Aperture calibration is based on the phenomena that the presence of low dense air or water in the rock matrix reduces CT numbers of voxels containing present voids and also can affect adjacent voxels. Considering a fractured rock along a cross section perpendicular to the fracture plane, the resulting density contrasts in most cases can be perceived as a more or less pronounced anomaly depending on fracture width. Thus, an aperture calibration method (MA method) was developed by Johns et al. (1993) assuming that all X-ray attenuation is conserved in the CT image and that local apertures can be derived by integrating the available density anomalies. Dispersion of X-ray attenuation and partial volume effects can cause an expansion of the anomaly over adjacent voxels that gather this "missing attenuation" (Johns et al., 1993) and, in particular





for larger fracture, represent a large portion of the entire anomaly. According to that, Johns et al. (1993) suggested a calibration-based linear relationship between aperture width and the integral of the full measured anomaly which was subsequently discussed and confirmed in several fracture aperture studies (Bertels et al., 2001; Heriawan and Koike, 2015; Huo and Benson, 2015; Keller, 1997; Ketcham et al., 2010; Van Geet and Swennen, 2001; Vandersteen et al., 2003;

Weerakone and Wong, 2010). Accordingly, the linearity between missing attenuation ($CT_{MA}$) and fracture aperture $a$ can be simply described as:

$$CT_{MA} = C \cdot a \tag{1}$$

where the constant C is given by the slope of the calibration line. Furthermore, $CT_{MA}$ is defined as

$$CT_{MA} = \sum_{i=1}^{N}(CT_{mat} - CT_i) \tag{2}$$

where $CT_{mat}$ describes an ideal value representing the rock matrix, $CT_i$ represents the CT number of the voxel along the cross

section affected by the missing attenuation and N localizes the considered voxel. In this study, $CT_{mat}$ is determined by averaging the single modes of all CT numbers at every pressure stage, assuming that the most frequent CT number dominates the matrix of the rock sample. Our results reveal a relatively high matrix number of $CT_{mat} = 1862$ HU due to the high density of the sandstone, which is in line with other CT-based sandstone studies (Akin and Kovscek, 2003; Huo et al., (2016); Vinegar et al., 1991). The MA method also allows the utilization of local $CT_{mat}$ values, which is potentially more

suitable when considering heterogeneous materials, but is rather applicable for studies focusing the fracture plane. Since the studied sandstone does not comprise significant variations in mineral density, a global $CT_{mat}$ is assumed to be sufficient.

Calibration with different spacers (0.19, 0.29, 0.41, 0.52 mm) within the fracture indicates a slope of the linear calibration line of $5890 \pm 38.3$ HU/mm (Figure 4). According to Eq. (1) and (2) apertures, therefore, can be calculated by:

$$a = \frac{\sum_{i=1}^{N}(CT_{mat} - CT_i)}{5890 \pm 38.3 \ HU/mm} \tag{3}$$

As a consequence, Eq. (3) (with N > 1) can be used to describe local apertures along the fracture; however, it is not practical to model the entire core sample. Typically, MA in voxels adjacent to the voxel containing the fracture depends on rock type

and aperture size (Huo et al., (2016)). Considering several cross sections through the fracture used here indicates that local apertures predominantly affect 2-3 voxels, where the vast majority of the attenuation is captured by the central voxel. Hence, assuming that the main information about fracture aperture is stored in one voxel, we define N = 1, which also benefits the calculation of apertures for every single voxel of the core sample. It should be mentioned that this method works well for the



aperture widths and voxel resolutions used here. However, depending on both, increasing aperture width and/or decreasing resolutions, the measured anomaly can distinctly extend over more than one voxel. This would cause artifactual apertures in the adjacent voxels actually representing the matrix and also would underestimate apertures in the voxels containing the fracture. In this study, most voxels containing the fracture range between 1700 and 1820 HU suggesting that calculated (and

summarized) apertures of each aperture cannot exceed 0.15 mm and most widely are significantly smaller than 0.1 mm. This corresponds to Huo and Benson (2015), who determined mean apertures between 0.025 to 0.031 mm. Negative apertures due to $CT_i > CT_{mat}$ as a result of the heterogeneity of the matrix or due to image noise are defined as "zero apertures" with a = 0 mm.

Although, this strategy does not describe the basic intention of the MA method, it provides a convenient solution to include data for the entire core and also enables integrations of detectable pores within the matrix. Hence, voxels representing sections of the matrix with enhanced porosity should reveal lower CT numbers ($CT_i < CT_{mat}$) so that these voxels are treated as equivalent apertures. These equivalent apertures represent the effective hydraulic diameter, which can be applied to characterize tubelike fluid flow in porous media (Debbas and Rumpf, 1966), and describes them as rectangular ducts

comprising the same area as the tubes (Janna, 2010).

Since simulating fluid flow does not work only with aperture information, each voxel has to be assigned to a potential hydraulic property. Assuming laminar flow and that fluid flow within the core is mainly controlled by fracture properties, the "cubic law" approach (Boussinesq, 1868; Snow, 1965) is chosen to derive appropriate local permeabilities ($K_{MA}$) based on

the MA aperture calibration, which is defined as:

$$K_{MA} = \frac{a^2}{12} \tag{4}$$

### 2.4 Simulation

Fluid flow within the processed CT scan image is simulated using the FlowDict module of the multidisciplinary, commercial

GeoDict® program package (Math2Market, Kaiserslautern, Germany), which has been developed to predict (microstructure-based) physical material properties (Pfrang et al., 2007). In the past, GeoDict was successfully applied to several studies with geoscientific background concerning fluid flow or particle tracking in fractured as well as porous media (Khan et al., 2012; Leu et al., 2014; Pudlo et al., 2014; Rücker et al., 2015; Schwarz and Enzmann, 2013). The utilization of the FlowDict module is based on three basic prerequisites: (1) a 3D voxel-based image of a permeable object, (2) an incompressible

Newtonian fluid and (3) experimental parameters such as mass flow and flow direction. FlowDict only supports two simulation types: Predicting the mean fluid velocity based on pressure drop boundary conditions and vice versa. Afterwards, entire sample permeability can be derived by applying Darcy's law (only for considering laminar flow).



In this study, fluid flow simulations are based on an explicit finite-volume (FV) method solving the Navier-Stokes-Brinkman equation according to Iliev and Laptev (2004). The governing equation comprises both, incompressible isothermal flow in pure fluid regions represented by the Navier-Stokes equation and flow in regions defined by hydraulic properties represented by the Brinkman extension of Darcy's law (Iliev and Laptev, 2004):

$$\underbrace{-\mu\Delta\,\vec{u} + (\rho\vec{u}\cdot\nabla)\vec{u}}_{Navier-Stokes} + \overbrace{\mu K_x^{-1}\cdot\vec{u} + \underline{\nabla P = 0}}^{Darcy's\ law} \tag{5}$$

$$\nabla\cdot\vec{u} = 0 \tag{6}$$

where μ is the fluid viscosity, ρ the fluid density, P represents fluid pressure and u is the three-dimensional velocity vector.

Furthermore, the reciprocal local permeability $K^{-1}_x$ represents locally assigned permeability based on the aperture calibration (x = MA), in voxels representing the solid matrix (x=solid) and further materials such as filter materials (x = filter). Equation (5) outlines the momentum conservation containing viscous forces, an advective acceleration term, the Brinkman extension to Darcy's law and the applied pressure gradient. Concurrently, continuity equation (Eq. (6)) describing the conservation of mass has to be valid. In order to solve the pressure-velocity relationship, a guess-and-correct procedure represented by the

Semi-Implicit Method for Pressure-Linked Equation (SIMPLE) algorithm (Patankar and Spalding, 1972) is implemented.

Fluid flow is simulated parallel to the core axis for pure water at 50°C with respective values for density (987.7 kg m$^{-3}$) and viscosity ($5.47 \times 10^{-4}$ Ns m$^{-2}$). The simulations compute the velocity field for the given pressure drop using periodic boundary conditions on the computational box and Dirichlet boundary conditions for the pressure. Pressure drops for each

stage of σ' are arbitrary defined by measured values obtained for a flow rate of 16 ml/min. $K_{solid}$ ($5.92 \times 10^{-19}$ m²) were obtained by MICP analysis (Sect. 2.1). In addition, in order to stabilize the simulation and to provide homogeneous in-/outflow as experimentally accomplished by filter plates, two artificial filter plates with $K_{filter} = 1 \times 10^{-10}$ m² are attached to the inlet and outlet according to the experimental setup (cf. Figure 3). Indeed, hydraulic properties of these filter plates are also included into flow simulations and associated permeability calculations; however, affectations can be assumed to be

negligible due to their proportionally small extent of the filter plates (194 × 194 × 5 voxels) compared to the sample dimensions (194 × 194 × 258 voxels).

Simulations are terminated by either reaching the accuracy criterion (ratio of current to former calculated permeability during iteration) of $1.5 \times 10^{-4}$ or exceeding $10^6$ iterations. Simulations are performed with a high-performance computer

(HPC) containing four Interlagos processors (64 cores) with 512 GB of total RAM. Dependent on available capacity (24 or 48 cores in this study) and aperture calibration, computing time of a single simulation ranges between two and four hours.



# 3 Results and discussion

## 3.1 Simulation

In this section, only the results of the simulations are discussed. For more detailed discussion and analysis on the experimental results, which particularly emphasize the hydraulic and geometric properties of the fracture as well as the

applicability of empirical models, we refer to Huo and Benson (2015). However, some of their findings are discussed in the context of the quantitative and qualitative simulation results presented in this study.

Qualitative results mainly highlight the visualization of fluid flow simulations that can be used for flow path analysis. On the other hand, quantitative results refer to fluid volumes for the different stress stages that can be directly validated by

corresponding core-flooding experiments and are compared to the effectiveness of other simulation approaches.

Visualizations of all simulation results indicate that principal fluid flow in the core is governed by a few preferential flow channels along the fracture plane that are partially interconnected (Figure 5). Differentiating fluid flow simulations for single stresses shows the expected stepwise closure of the fracture and containing channels. This becomes clear for considering

changes in local fluid flow with changes in effective stress. Applying loading conditions indicate a significant decrease in connectivity and local permeabilities occurring predominantly within most parts of the fracture. In reverse, the same is observed for unloading conditions. This behavior is in line with an increasing (decreasing) percentage of contact areas and an increase (decrease) of smaller apertures due to loading (unloading) presented by Huo and Benson (2015). Furthermore, applying unloading conditions includes reincreasing of connectivity and permeability due to the opening of several channels

and branches. However, some channels and, especially branches, remain closed or disconnected as a result of irreversible deformation (Figure 5). Hence, it can be assumed that changes in fracture connectivity can be directly related to the strength of the present asperities (Huo and Benson, 2015; Pippan and Gumbsch, 2011). Accordingly, plastic deformation and brittle failures along asperity tips cause irreversible changes that are associated with the permanent local closings and constrictions as also depicted by Figure 5, while elastic deformation can be assumed to be most widely reversible and facilitates local

reopenings.

Furthermore, additionally enhanced fluid flow is observed in single parts within the matrix (Figure 6). However, the overall matrix permeability can be assumed to be rather low which is also clarified by carefully examining the propagation of the pressure fields. Concurrently, propagation of the pressure fields also reinforces the assumption that major fluid flow occurs

along the fracture. Unfortunately, core permeability is derived by solving the Navier-Stokes-Brinkman equation for the measured pressure drop over the entire sample (as stated in Sect. 2.4) which prevents the quantification of the matrix permeability. Hence, precedent core-flooding experiments with an unfractured core sample should be considered in future experiments. Furthermore, the experimental procedure did not allow the utilization of appropriate contrast agents during CT





experiments to validate flow paths. According to these limitations, distinct flow patterns along areas of higher matrix porosity should be treated with caution, since actual sizes, numbers and connectivity of pores within a single voxel are not explicitly known. Thus, calibrated equivalent apertures and derived permeabilities per voxel only represent a simplified approximation of the actual geometry and connectivity of present pores. Nevertheless, the simulated permeability anisotropy

is not unlikely, since the sandstone sample features conspicuous laminations as stated previously, indicating significant porosity heterogeneities. Simulated permeability anisotropy also agrees with other CT-based studies examining laminated sandstone samples (Clavaud et al., 2008; Grader et al., 2013; Karpyn et al., 2009). Embedded low-density minerals mimicking "non-zero apertures" as described for coarser granite samples by Watanabe et al. (2011) most widely can be excepted for the finegrained sandstone sample. As inferred from visualizations, a (hypothetical) fluid volume initially

appears to flow along the channel network of the fracture until it is partially split by an intersecting high-porosity layer, where a portion follows the dipping of lamination in outflow direction.

Comparing absolute changes in fluid flow (Figure 6) due to loading between the lowest (0.7 MPa) and the highest (22.1 MPa) pressure stage indicates that most changes within the sample occur along the fracture plane. Indeed, there are minor

changes within the matrix that appear to be caused by processes within the matrix; however, simulations indicate that these changes hardly affect the fluid flow within the matrix.

In contrast to qualitative analysis, quantitative results of the simulation significantly differ from expected results (Figure 7a). Although, there are significant deviations, quantitative results exhibit several characteristics also observed in the experiment

(Huo and Benson, 2015) such as (weakened) hysteretic behavior, discernible decreases of permeability with increasing pressure and subsequent reactivation of fluid paths under unloading conditions. A first simple (straightforward) simulation is based on an aperture calibration applying one global CT number for the matrix material ($CT_{mat}$ =1862.6 HU) as introduced in Sect. 2.3. Additionally, two further simulations are based on modified $CT_{mat}$ values. For the latter, the standard deviation of the voxels is considered. Commonly, image quality of CT scans can be affected by random noise caused by imprecisions

during CT image reconstruction (Huo et al., (2016); Pini et al., 2012) or by the reduction of X-ray intensity due to the presence of metal filters to reduce beam hardening (Ketcham and Carlson, 2001; Nakashima and Nakano, 2014; Watanabe et al., 2011). As a result of the induced noise, Huo et al., (2016) ascertained a mean standard deviation of ± 17.7 HU for each of the voxels. Indeed, the authors accomplished a further reduction of this uncertainty (± 7.9 HU) for their purposes by generalizing significantly lower $CT_{mat}$ values (~1800 HU) over single transects through the fracture plane containing six

matrix voxels. However, in this study, applying ± 17.7 HU is appropriate due to the voxel-wise calibration approach of the CT images. Thus, two "worst-case" scenarios are considered by implementing the terms $CT_{mat}$ ± 17.7 HU as input parameters for model calibration of these scenarios to account for uncertainties in $CT_{mat}$.



Considering the simple modelling approach, the simulated permeabilities are in the range of minimum and maximum measured permeabilities. However, the permeabilities are clearly underestimated (up to a factor of 0.3) at lower σ' and significantly overestimated (up to a factor of 7.1) at higher σ'. As a result, slopes of stress-dependent permeability curve diminishes for lower σ' under loading as well as unloading conditions, which consequently contributes to a declined

hysteresis area ($6.71 \times 10^{-13}$ MPa × m²) compared to experimental results ($3.17 \times 10^{-12}$ MPa × m²). This reduction of hysteresis area indicates that intensity of fracture closing derived from CT images appears to be depressed. Possible explanations for this observation are discussed hereinafter.

Comparing the deviation between predicted and observed results indicates that there appears to be a systematic error

affecting the simulation results (Figure 7b). Of course, this could be caused by uncertainties of the pressure drop measurements during the experiment. However, permeabilities calculated by considering pressure drop uncertainties indicate deviation by a maximum factor of 1.2 and a minimum factor of 0.9 at the lowest loading stage (Huo and Benson, 2015). Most factors of other pressure stages are about 1.0 so that this uncertainty seems to be negligible. Counterchecking of the model input data and boundary conditions showed that the error of the simulation mainly relies on the aperture calibration.

For that reason, we tested the sensitivity of the simulation focusing constants (slope and $CT_{mat}$) and shape of the calibration line.

By varying the slope (5980.2 HU/mm, according to Eq. (3)) between ± 1 to 5 % hardly reveals any effect on the hysteresis except a very slight shift along the ordinate. This should be explained by very low apertures within the fracture, as

represented by mean apertures between 0.025 and 0.031 mm as shown by Huo and Benson (2015), so that changes within the slope hardly affect apertures at the considered scale.

Furthermore, in order to be valid and as introduced in Sect. 2.3, the MA method requires a linearity of the calibration line and which also justifies the extrapolation of apertures as performed here. On the other hand, Mazumder et al. (2006)

described a nonlinear relationship between apertures and $CT_{MA}$ for fractured coal samples using an optimized MA approach. The derived calibration line indicates an increase of the slope with decreasing apertures, which would cause significant deviations in aperture calibration compared to a linear approach. However, available calibration data of our study are rather linear, so that simulations that are based on nonlinear regressions do not significantly differ from the linear approaches.

An additional factor that can contribute a major part to the simulated deviations is represented by $CT_{mat}$ describing an averaged threshold value for the matrix material. Since significant material heterogeneities appear to be negligible for the sandstone, we focus here on the sensitivity of the simulations caused by uncertainties in $CT_{mat}$. Thus, $CT_{mat}$ is varied by the uncertainties provided by random noise by including the averaged standard deviation (± 17.7 HU). Adding the standard deviation to $CT_{mat}$ + 17.7 HU indicates that simulated results at lower σ' are shifted towards the experimental data while





results at higher σ' significantly overestimate the measurements. The opposite happens for $CT_{mat} - 17.7$ HU where simulations rather approximate measurements at higher σ' and strongly underestimate measurements at lower σ'. According to Eq. (3) and Figure 4 this is obvious since higher $CT_{mat}$ causes the calibration of larger apertures and, therefore, partially open new flow channels by ascribing apertures that are zero for non-adjusted $CT_{mat}$. Vice versa, non-adjusted channels are

constricted or closed by slightly tighter apertures applying a lower $CT_{mat}$. Hence, the simulation results are highly sensitive to changes in $CT_{mat}$. Modifying $CT_{mat}$ just by +17.7 HU increases simulated permeabilities up to a mean factor of 2.6 ± 0.1. In contrast, reducing $CT_{mat}$ by -17.7 HU diminishes simulation results up to a mean factor of 0.3 ± 0.02. Hence, the simulated results that contain the standard deviation of the CT scan are able to cover the full range of experimental values. However, the latter also clearly demonstrate that the used simulation is unable to explicitly reproduce the measured

permeability hysteresis.

So, theoretically, the measured permeabilities could be fitted by varying $CT_{mat}$ within the range provided by the standard deviation. Comparing the experiment and the three simulations, one may assume stress-dependency of $CT_{mat}$, so that a correction factor could fit the simulations. However, a correction factor based on the observed trend would imply a decrease

of $CT_{mat}$ with increasing σ' suggesting a decrease of numerical material density. In fact, increasing of σ' should cause an increase in density. Hence, $CT_{mat}$ typically should increase with increasing σ', which would cause extra discrepancies of the simulated permeabilities. Actually, no systematic increasing or decreasing is observed for $CT_{mat}$ at single σ', neither for the modes of the entire core sample or for selected elementary volumes representing only the matrix. Instead, single $CT_{mat}$ rather oscillates around their average with amplitudes below the residual noise (± 17.7 HU). Thus, applying a single average $CT_{mat}$

is supposed to be the most satisfactory solution for our purposes.

Accordingly, the most reasonable explanation for the deviations the simple simulation approach (with $CT_{mat} = 1862.6$ HU) from the experimental hysteresis is the presence of potential sub-grid scale features. This means that variations in the fracture topography most likely are significantly below the CT resolution ($0.5 \times 0.5 \times 1.0$ mm³) so that the actual roughness

cannot be captured accurately. Thus, the resolution-caused generalization of these features per voxel underestimates experimental permeabilities at lower stresses due to a reduction in actual (sub-grid-scale) connectivity. While at higher stresses the permeability is overestimated by suggesting flow paths that actually are clogged by (sub-grid scale) contact areas. Therefore, the transferred fracture roughness represents only an approximation of a finer and more complex flow pattern within smooth fracture where connectivity can be assumed to play a significant role.

**3.2 Comparison**

As stated above (Table 1), several authors have published stress-dependent fluid flow studies based on variously derived fracture geometries and numerical approaches. Until now, medical CT data and, thus, most untreated input data solely were



applied by Watanabe et al. (2011) using an aluminum core-holder for core-flooding experiments on tensile single-fractured (SF for single fracture) and naturally double-fractured (MF for multiple fractures) granites. In Figure 8 simulated permeabilities resulting from their approach are compared to the results of this study (only loading conditions are shown). As one might expect, measured permeabilities of the four core-flooding experiments decrease with increasing σ' (Figure 8a).

Although, both rock types, Zenifim Sandstone and granites, reveal similar matrix permeabilities ($10^{-18}$ to $10^{-19}$ m²), fluid flow experiments with the fractured sandstone obtain significant higher permeabilities than granite experiments (up to 3 orders of magnitude). Typically, mean apertures of mated tensile fractures in Inada granite should decrease from 0.071 to 0.065 mm applying loading from 10 to 100 MPa (Watanabe et al., 2008). Mean apertures of the sandstone sample are significantly lower and range between 0.031 and 0.024 mm (Huo and Benson, 2015). Nevertheless, with increasing stresses, hydraulic

apertures of the granite fracture decrease from 0.009 to 0.003 mm (Watanabe et al., 2008) while hydraulic apertures of the sandstone fracture decrease from approximately 0.030 to 0.005 mm (Huo and Benson, 2015). This significant hydraulic differences of the samples appear to be caused by higher tortuosity described by (Tsang, 1984) due to a higher roughness of the tensile granite fractures as compared with the smooth sandstone fracture. In addition, this dependency between fluid flow and tortuosity is reinforced by considering CFD-based flow visualizations of the granite sample indicating a highly tortuous

network of flow channels along the fracture (Watanabe et al., 2011).

In order to clarify the effectiveness of the compared simulation approaches, discrepancies between simulated and experimental permeabilities of each sample are plotted as factors against the normalized effective stress in Figure 8b. It can be seen that simulated permeabilities of the SF sample are 5 (at 5 MPa) to 116 times (at 50 MPa) higher than predicted by

the experimental data, while simulations based on the MF sample overestimate experimental results by a factor of 30 (at 5 MPa) and 106 (at 50 MPa). Overestimations are supposed to be primarily caused by image noise due to the core-holder causing "non-zero apertures" and, in consequence, extended flow paths within the fracture and the matrix (Watanabe et al., 2011). Considering the three simulation scenarios conducted within the scope of this study, experimental permeabilities are over- or underestimated ranging between a minimum factor of 0.1 and a maximum factor of 20. Considering only the simple

simulation approach (with $CT_{mat} = 1862.6$ HU) numerical results show a maximum overestimation by a factor of 7.1 and a minimum underestimation by a factor of 0.3. Accordingly, the MA method introduced in this study appears to reveal slightly better simulation results than Watanabe et al. (2011), although the considered fracture is significantly smaller which indicates an effective noise reduction by applying multiple scans. However, in a subsequent study Watanabe et al. (2013) improved their approach (Figure 8) with an adjusted experimental setup on a granite containing a single tensile fracture (SF

PEEK) by using a carbon fiber reinforced polyetheretherketone (CFR PEEK) core-holder. Applying the PEEK core-holder significantly reduced the image noise and revealed nearly concurrent experimental and simulated results.

Accordingly, the introduced method represents a further approximating step for successful CFD simulations, but also exposes current limitations. However, by focussing on our own experiences and including the pioneering research by



Watanabe et al. (2011) and Watanabe et al. (2013), we provide the following recommendations for future medical CT-based fracture flow studies.

- The determination of more detailed pore data of the matrix, especially when considering heterogeneous porous rocks, by introducing additional MICP measurements and technologies such as micro-CT, Environmental Scanning Electron Microscopy (ESEM) or Nuclear Magnetic Resonance (NMR) spectrometry. Additionally, an upscaling of micro-CT data to core scale and coupling this matrix with a medical CT based aperture distribution of the fracture would provide more realistic pore-geometry data for simulating fracture-matrix flow interaction.

- Before fracture flow experiments start, some "blank tests" should be considered. Core-flooding experiments of the unfractured core-sample can provide additional information about the actual hydraulic properties of the matrix. Furthermore, scans of the unstressed fractured sample with a PEEK core-holder (to reduce the beam hardening effect) can be used to detect image noise, which is caused by the metal core-holder used in common core-flooding experiments as presented by Watanabe et al., 2013. In this study, image noise cannot be explicitly assigned to the core-holder and is most widely due to the imperfections of the CT image reconstruction as also stated for equivalent experimental setups (Huo et al., (2016); Pini et al., 2012). However, an additional affectation due to the core-holder cannot exclude so that a "blank test" would be very beneficial. Since both mentioned noise sources create random noise in the CT image both quality improvement methods, the PEEK core-holder and the multiple scan method, should be tested for their effectiveness. Ideally, both methods should be combined to diminish affectations of both possible noise sources. Additionally, this study is based on low voxel resolutions ($0.5 \times 0.5 \times 1.0$ mm³) generalizing the area, which is covered by the voxel, to a single aperture value. Thus, high-resolution fracture measurements such as laser scanning or profilometer measurements should be helpful to detect the effect of generalized fracture morphologies within the voxel-scale by comparison with unstressed "blank test" data.

- However, PEEK core-holder technology was only successfully applied for maximum confining pressure between 36 and 50 MPa at room temperatures (23°C) (Ito et al., 2013; Watanabe et al., 2013) and, thus, is a useful option for experiments focussing similar conditions. However, the strength of PEEK strongly depends on temperature (Searle and Pfeiffer, 1985) and the chemical composition of the fluid used for the experiment (Pritchard, 1994) and, thus, appears to be challenging for experiments that simulate reservoir conditions or processes in the deeper Earth's crust.

- Typically the aperture calibration used in this study suggests that MA and, thus, apertures only can be determined perpendicular to the fracture plane rather favouring 2D aperture distributions. Despite that, we found a good transferability of present apertures to a 3D problem favoured by the tightness of the studied fracture with mean apertures between 0.031 and 0.024 mm, where the fracture-caused anomaly predominantly affects the voxel




containing the fracture. However, with increasing fracture width it becomes more likely that the anomaly also affects adjacent voxels and the aperture calibration becomes invalid. Thus, the applicability of the introduced method should be validated for additional scenarios: (1) For different fracture widths where apertures must be smaller than the x-y resolution of the CT scan. (2) For different fracture types such as smooth and rough fractures or tensile and shear fractures. (3) For different materials highlighting the influence of different heterogeneities caused by the mineralogical composition or porosity. (4) For different applicable x-y resolutions provided by medical CTs (e.g. $0.2 \times 0.2$ mm² or $0.3 \times 0.3$ mm²).

- However, most of the suggested scenarios depend on each other and should be critically considered. In addition, a similar procedure is recommended for the PH method used by Watanabe et al. (2011) to underline the scope of applications and limitations of both methods.

- In this study, aperture calibration is based on the general assumption of a linear relationship between MA and aperture width (e.g. Bertels et al., 2001; Keller, 1997; Van Geet and Swennen, 2001) and, thus, on a crude spacer calibration (with 4 spacers $\geq 0.2$ mm). However, Mazumder et al. (2006) found a significant nonlinear relationship by applying an optimized MA approach to minimize affectations due to heterogeneities of their coal sample. They found that the slope of the line significantly increases with decreasing apertures (< 0.25 mm). Therefore, future studies should focus a more accurate spacer calibration, particularly with spacers < 0.1 mm, which accords to many apertures under confining pressure.

In order to validate qualitative fluid flow, new contrast agents should be introduced. Conventionally used iodine was shown to cause beam hardening effects that exacerbate the interpretation flow paths within the fracture (Watanabe et al., 2011). Meanwhile, sodium polytungstate ($Na_6H_2W_{12}O_{40}$) was proposed as a promising contrast agent for hydrological CT experiments by significantly reducing the undesirable beam hardening effect (Nakashima, 2013; Nakashima and Nakano, 2014). Alternatively, further technologies such as positron emission tomography (PET) could be applied (Fernø et al., 2015; Kulenkampff et al., 2008).

**4 Conclusion**

A novel method to simulated fluid flow in a fractured porous core sample under loading and unloading based medical CT measurements was introduced. Simulation results reveal qualitative plausibility, but also reveal shortcomings considering quantitative results. Indeed, the proposed method is able to approximate experimentally derived permeability data; however, it merely indicates the stress-dependency of fracture permeability.



Qualitative results reveal satisfactory accordance with the well-established flow channeling approach indicating that major flow is governed by several preferential flow paths along the fracture with less pronounced tortuosity. The simulations reproduced temporal and permanent closing of some flow channels due to increasing effective stress causing significant changes in connectivity and associated permeability. Furthermore, the applied simulation strategy indicate fracture-matrix

interaction and permeability anisotropy. The anisotropy appears to be caused by laminated porosity heterogeneities within the matrix.

Despite the quantitative deviations, the simulated permeabilities indicate stress-dependency of the sample represented by a slight decrease in permeability with increasing effective stress and even imply hysteretic behavior. The deviations appear to

be mainly caused by resolution-caused limitations which prevent an accurate capturing of sub-grid scale features which affect the reproduction of actual connectivity playing an important role in smooth fractures. Furthermore, the simulation is very sensitive to the choice of an adequate threshold value $CT_{mat}$ (1862.6 HU in this study). Small deviations from the ideal $CT_{mat}$ ($\pm17.7$ HU in this study) can cause enormous changes in simulated permeability by up to a factor of $2.6 \pm 0.1$. Thus, $CT_{mat}$ has to be defined with caution and can cause additional problems for rocks with significant mineralogical

heterogeneities.

A comparison with a similar study by Watanabe et al. (2011), who used an equivalent experimental setup, but different numerical approach (based on the PH method), indicate similar invalidity of simulated permeabilities, however enables the generation of a recommendation list for future research including: (a) Extensive, preliminary porosity studies and "blank

tests" with the unfractured core to expose more detailed hydraulic properties of the sample. (b) "Blank tests" by applying a PEEK core-holder should reveal the influence of image noise when using a metal core-holder for the experiments. Additionally, high-resolution measurement techniques should indicate possible effects of small-scale (<0.5 mm) fracture morphologies that are generalized due to the CT resolution. (c) The development of PEEK core-holders that are able to resist experiments under reservoir conditions. (d) A direct comparison of the applied MA and PH method to consider the merits

and demerits of both approaches focussing on different fracture widths and types, different matrix materials and CT resolutions. (e) A more detailed analysis of the aperture calibration for the MA method by focussing smaller apertures (< 0.1 mm). (f) The development of new contrast agents or utilization of alternative technologies to validate qualitative fluid flow within the fracture.

**Data availability**

Since the size of the underlying data is too large for an upload, the authors encourage interested readers to contact the co-authors. Raw CT data can be obtained from DH (dhuosu@gmail.com). Processed CT data and simulation results (e.g. flow and pressure patterns) are stored on a server at the University of Mainz (Contact: enzmann@uni-mainz.de)





## Acknowledgement

This study was mainly carried out within the framework of the Helmholtz Association of German Research Centres (HGF) portfolio project 'Geoenergy' and is part of the comprised reservoir engineering cluster. In addition, we acknowledge the postdoctoral grant to JOS, which was funded within the frame-work of DGMK (German Society for Petroleum and Coal

Science and Technology) research project 718 "Mineral Vein Dynamics Modelling". The latter is funded by the companies ExxonMobil Production Deutschland GmbH, GDF SUEZ E&P Deutschland GmbH, DEA Deutsche Erdoel AG and Wintershall Holding GmbH, within the basic research program of the WEG Wirtschaftsverband Erdöl- und Erdgasgewinnung e.V. The authors also want to thank Rani Calvo from the Geological Survey of Israel for providing the Zenifim sandstone sample used in this study. In particular, we thank to Math2Market providing the GeoDict software

package for fluid flow simulations. Furthermore, the authors are grateful to the Karlsruhe Institute of Technology (KIT), whose policy relating to open access journals facilitates financial support.

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




**Tables**

Table 1: Numerical simulations of stress-dependent, single-phase fracture flow considering the sample and fracture type, the
5   method to reproduce fracture apertures, the dimensions and simulation methods.

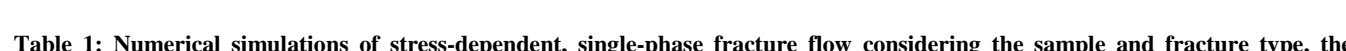

| References | Core sample (length/diameter) | Fracture type | Source of model geometry (Aperture translation method) | Volume element dimension (represented fracture area) | Dimen-sions | Simulation type (Solver) |
|---|---|---|---|---|---|---|
| Pyrak-Nolte and Morris (2000) | Granites (literature data) | Artificial | Fractal aperture distributions (Stratified Percolation method) | $300 \times 300$ cubic grid blocks ($100 \times 100$ mm²) | 2.5D | Network model (Hardy-Cross method) |
| Kim et al. (2003) | Granite (16.4 cm/14.0 cm) | Tensile (mated and offset) | Profilometer measurements | $0.05 \times 0.05$ mm $\times >0.008$ mm³ (111 to 116 $\times$ 128 mm²) | 3D | LB (Navier-Stokes) |
| Dicman et al. (2004) | Sandstone (2.54 cm/5.99 cm) | Tensile | Average aperture (Stochastic aperture distribution map) | $1 \times 1 \times 1$ grid block ($10 \times 10 \times 15$ grid blocks) | 3D | FD (Local cubic law) |
| Watanabe et al. (2008) | Granite (15.0 cm/10.0 cm) | Tensile (mated and offset) | Laser-scanning equipment | $0.25 \times 0.25$ mm² (sample scale) | 2.5D | FD (Local cubic law) |
| Nemoto et al. (2009) | Granite (15.0 cm/9.5 cm) | Shear fracture for experiment, tensile (offset) for simulation | Laser-scanning equipment and thin film technique | $0.25 \times 0.25$ mm² ($50 \times 50$ mm²) | 2.5D | FD (Local cubic law) |
| Liu et al. (2010) | Coal and Sand-stone (10.0 cm/5.0 cm) | Shear fracture | Conceptual side view | $100 \times 25$ four-node mesh (none) | 2D | FE (Fluid/solid coupling model) |
| Watanabe et al. (2011); Watanabe et al. (2013) | Granite (15.0 cm/10.0 cm) | Tensile (mated) and natural (double fractured) | Medical CT scanner (Peak Height method) | $0.35 \times 0.35 \times 0.50$ mm³ (sample scale) | 3D | FD (Local cubic law) |
| Indraratna et al. (2015) | Sandstone (11.4 cm/5.4 cm) | Tensile (mated) | Laser-scanning equipment | $1.0 \times 1.0$ mm² (sample scale) | 2.5D | FV (Navier-Stokes) |
| Current | Sandstone (6.7 cm/5.0 cm) | Saw cut (smooth, mated) | Medical CT scanner (Missing Attenuation method) | $0.50 \times 0.50 \times 1.00$ mm³ (sample scale) | 3D | FV (Navier-Stokes-Brinkman) |



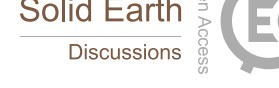

**Figures**

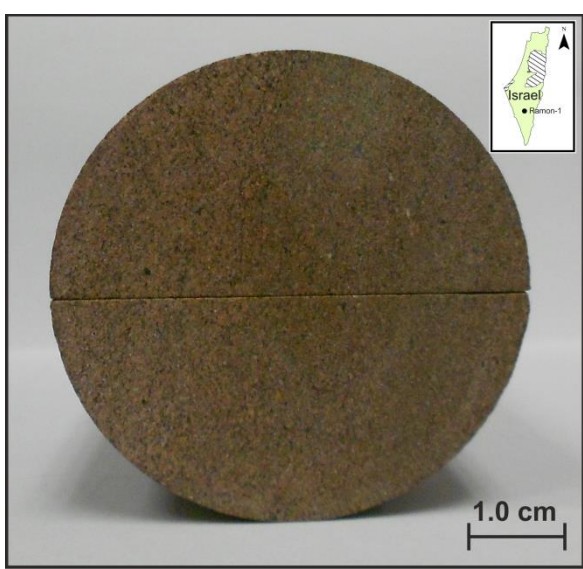

5   **Figure 1: Top view of the fractured Zenifim Sandstone with a diameter of 5.0 cm and a length of 6.7 cm.**

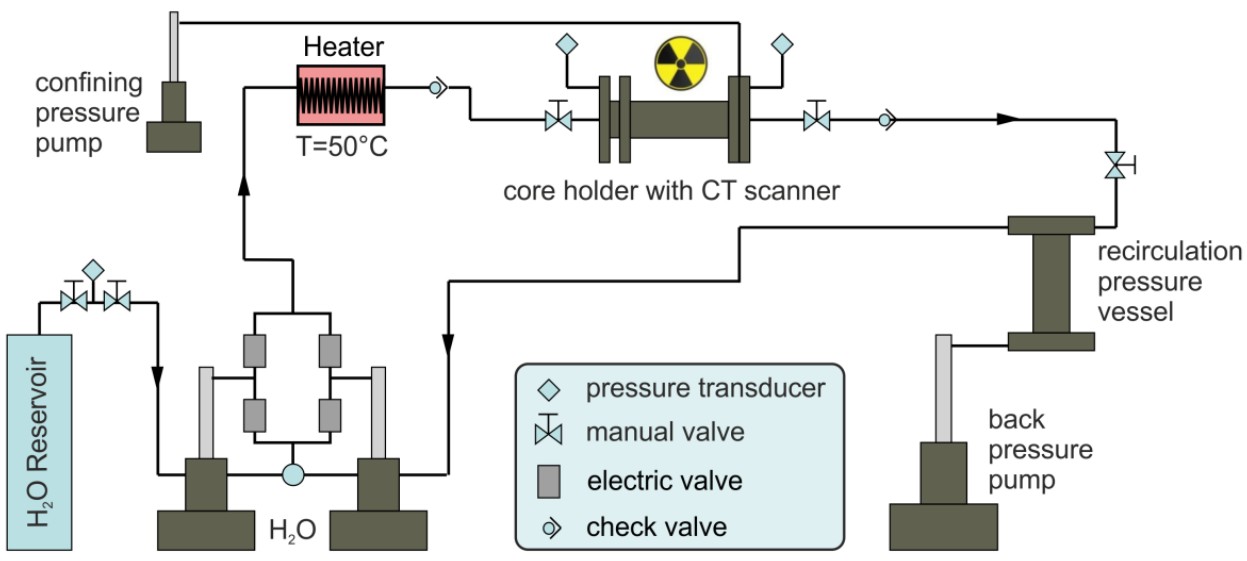

**Figure 2: Schematic setup of the core-flooding apparatus for single-phase (H₂O) flow experiments (after Huo and Benson, 2015).**



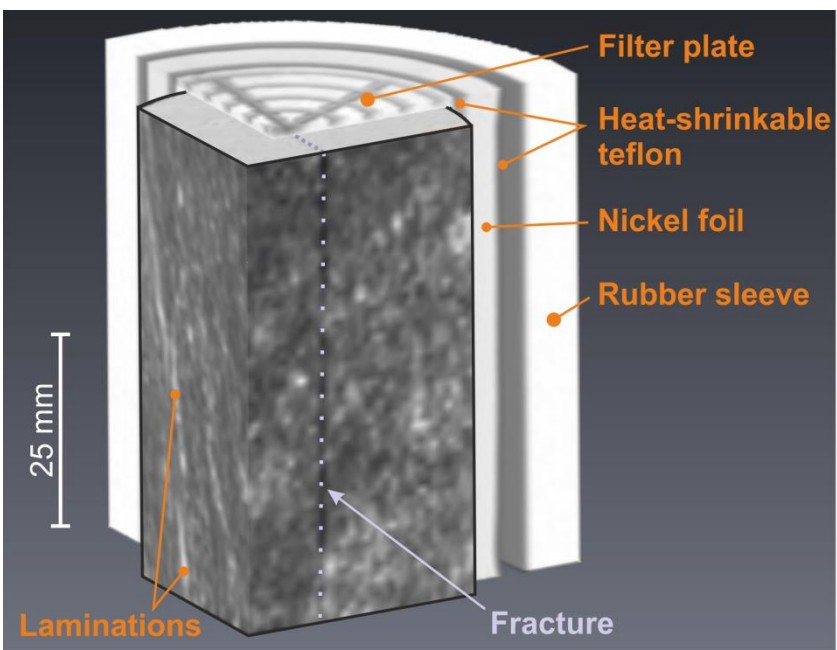

**Figure 3: The Computed tomography (CT) image of the rock sample with a rescaled voxel dimension ($0.25 \times 0.25 \times 0.25$ mm³). The upper filter plate, the units of the core-holder and lamination of the sample are highlighted in orange. The fracture is indicated by a dotted purple line.**

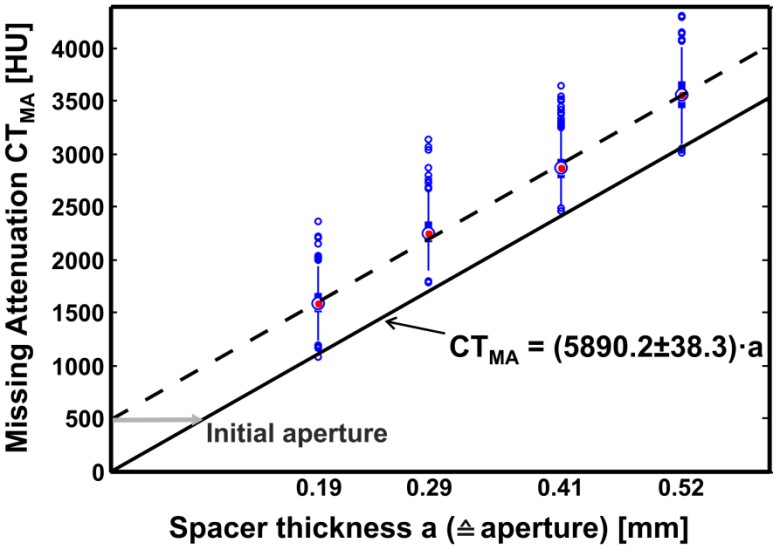

**Figure 4: Calibration based on a total of 380 points at each spacer in order to apply the missing attenuation method. The circles with the red dot represent the medians for every spacer while smaller circles without dot are outliers. The thin and thick blue lines show 25 % confidence intervals and 75 % respectively. The dashed line corresponds to the regression line of all calibration points.**
10    **The solid line corresponds to the adjusted regression line used for further aperture calculations by neglecting an apparent aperture (0.08 mm) caused by the fracture roughness (after Huo et al., 2016).**



**Figure 5: Changes in 3D fluid flow only along the fracture plane due to closing during loading conditions ( from 0.7 MPa to 22.1 MPa) due to reopening caused by unloading conditions (from 22.1 MPa to 0.7 MPa) by applying forward modelling with an EFV method solving Navier-Stokes-Brinkman. Fluid flow is visualized by fluid velocity per voxel normalized by the maximum fluid velocity at the given pressure stage. Fluid velocity increases from red to yellow. Velocity fields of the initial and residual pressure stage (0.7 MPa) indicate permanent local closings along the fracture. A schematic perspective is included for better 3D orientation.**





**Figure 6: Changes in 3D fluid flow sample due to closing (A and B) during loading conditions (from 0.7 MPa to 22.1 MPa) by applying forward modelling with an EFV method solving Navier-Stokes-Brinkman. Flow visualizations comprise fracture as well as matrix flow. Fluid flow is visualized by fluid velocity per voxel normalized by the maximum fluid velocity at the given pressure stage. Fluid velocity increases from purple to red. Absolute changes in fluid flow (|A-B|) are shown in the lower part. Most changes occur within the fracture. Absolute changes increase from purple to red. A schematic perspective is included for better 3D orientation.**





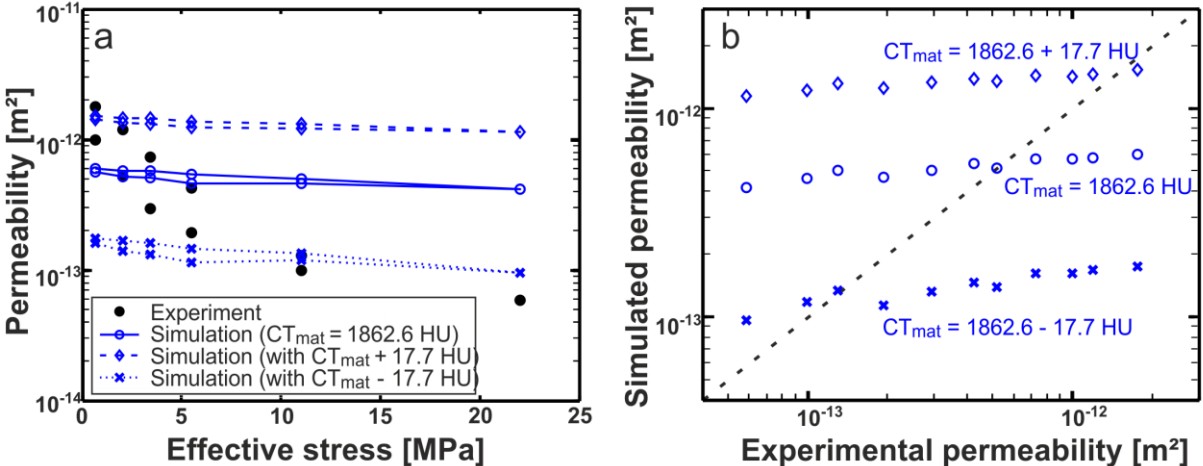

**Figure 7: (a)** Measured and simulated permeabilities of the entire core-sample versus effective stress. Simulation results are obtained by forward modelling (cf. Eq.3) and by changing $CT_{mat}$ according to the mean standard deviation of the CT measurements (cf. Eq.7) and **(b)** corresponding deviations between experimental and simulated permeabilities indicating a systematic error.

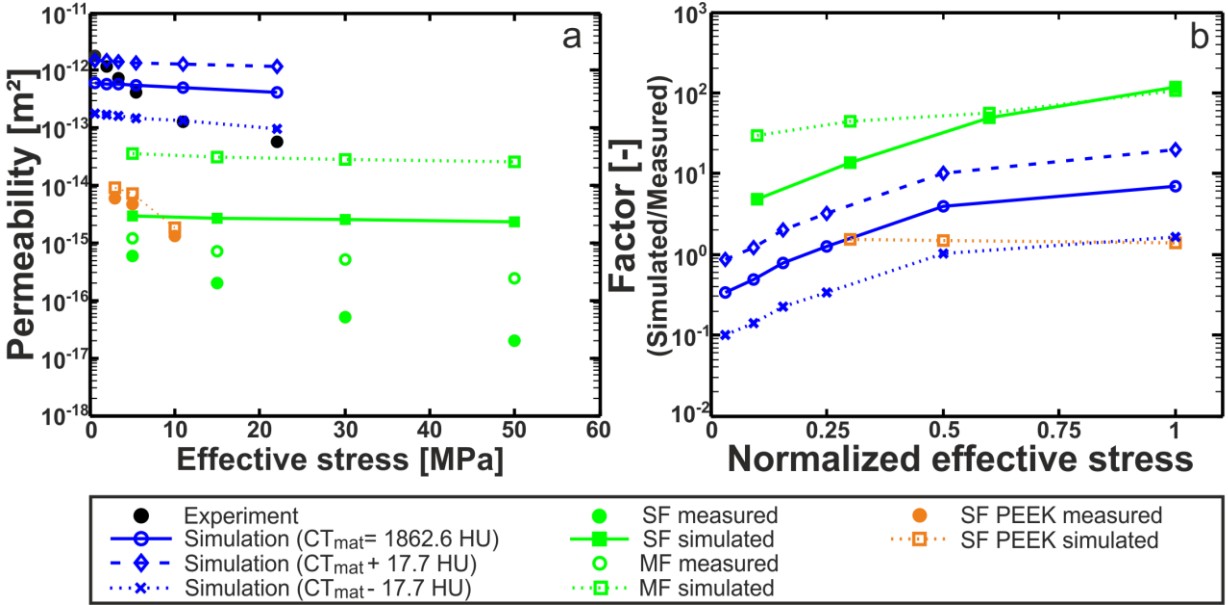

**Figure 8:** Comparison of experimental and simulated data of the sandstone sample with results obtained from the literature using CT technology to realize simulations (Watanabe et al., 2011; Watanabe et al., 2013). Literature data are based on a three granite core samples with a single, tensile fracture (SF), multiple, natural fractures (MF) and a single, tensile fracture measured by using an improved core-holder (SF PEEK). Furthermore, the effectiveness of the simulations is shown in **(b)** as factors, describing the discrepancy between each simulated and corresponding experimental permeability value, versus the normalized effective stresses depending on the highest stress stage applied during the associated experiment.