# Peer review of "Simulating stress-dependent fluid flow in a fractured core sample using real-time X-ray CT data"

_Solid Earth, 2016_

## Referee Comment (RC1) · Anonymous Referee #1 · 22 Mar 2016

Overall the approach, description and methods presented in the manuscript "Simulating stress-dependent fluid flow in a fractured core samples using real-time X-ray CT data" submitted to Solid Earth Discussions is good. I would recommend the following minor changes to enhance the presentation and discussion of this work.

Page 5, Line 9, if you could add the permeability of the rock matrix in mD that would save a reader from having to think about the conversion. This would be nice elsewhere in the document as well.

Page 6, lines 17 and 18 discussing the conversion of the original voxel dimension to the isotropic voxel size. This is unclear to me. If a 0.5 x 0.5 x 1 mm^3 voxel is converted to four (4) 0.25 x 0.25 x 0.25 mm^3 voxels there is a loss of volume somewhere I think.

I think the authors mean for this to be 16 voxels, but I am unsure on the conversion technique in general and would like for some more discussion on this process to be added to the text so that it is clear.

Page 6, line 27, 'dense' should be 'density' I think.

The use of the missing attenuation approach in general. Since it's stated on page 7, line 32 that N = 1 for the MA calculations then this isn't really a summation of the the HU in the aperture. The authors acknowledge this later on in the text (Page 8, line 10), but then double back in the conclusions and discussion to say that MA is good for identifying the aperture values. That just doesn't sit very well with me and more clarification on this when describing the results would be preferred. Something like "the modified MA using the primary voxel within the fracture showed good results"

I'm not as convinced as the authors are that conversion to a PEEK core holder would improve results of the simulation so much. But I'm willing to let that slide, as it would reduce CT noise and improve results somewhat.

I think the largest issue with the simulations is the constant matrix permeability. The authors discuss this several times and attempt to correct for this by modifying the matrix permeability +/- one standard deviation of the HU (Fig 7b). I think the variability in the matrix would need to be accounted for by modifying the voxel permeability in the matrix to match high HU voxels with very low (zero) permeability and low HU voxels to higher permeability values. But I have no idea if GeoDict could actually handle this level of complexity. If it could, I think it would benefit your analysis and fit to the experimental data.

Overall, I think the manuscript is well written, identifies some issues with this type of analysis that are worth publishing, and is a good stepping stone to improved measurements and simulations of flow in fractures. I recommend publication with the minor issues listed above corrected to the best ability of the authors.

Thank you!

---

## Referee Comment (RC2) · Anonymous Referee #2 · 12 Apr 2016

The objective of the study is to evaluate a methodology to characterize the flow path geometry in a fracture as a function of stress history using medical CT images of the fracture under varying states of stress. The development of methodologies for digital rock physics workflows remains an active area of research and the investigation. However, the investigation of methodology presented here is incomplete, a local instead of global analysis of the uncertainty in fracture aperture estimation should be conducted. I have one fairly major concern regarding the methodology, and analysis of the methodology. Following this I also have outlined further comments/questions that should be addressed. As it stands this manuscript offers little in the way of novelty, and there are major issues with the methodology.

[Figure]

Page 3 lines 23-24, Page 6 lines 28-32, Page 7 lines 20-21: There is a major misconception regarding CT numbers throughout the manuscript, X-ray attenuation is a function of density and apparent atomic number (this is why contrasting agents are often employed!). The variation of the CT numbers for the sample shown is not only a function of density, but also composition. The variation of the CT numbers being a function of density and composition can have a significant effect on the estimation of the subvoxel fracture aperture.

I do not believe the methods used by the authors to extract the fracture geometry from the CT images have been properly assessed. Specifically, the effect a locally varying matrix CT number has on estimated subvoxel fracture apertures. The fracture apertures of interest are far smaller than the voxel dimensions. Therefore, as outlined by the authors, the apertures are instead estimated from the CT numbers of the voxels containing the fracture (CT(i)). To estimate the local aperture the authors use a method that requires assumptions that may be significantly erroneous; Equations (1) and (2) summarize the MA method,

Local Aperture = 1/C*CT(MA)=1/C*(CT(mat)-CT(i)).

The underlying conception of the MA method used by the authors assumes the following weighted volume average contributes to the measured CT(i),

CT(i)=(VolumeLocalMatrix/voxelVolume)*CT(local matrix) + (VolumeWater/voxelVolume)*CT(water),

where the voxel volume and CT(water) are constants, and CT(local matrix) is the CT number of the matrix within that voxel, CT(i). Only CT(i) is known, therefore, unless CT(local matrix) is also known, the volume of water occupying the voxel – and subsequently the estimated aperture of the fracture – cannot be determined. It is apparent from the CT image in figure 3 that the matrix CT numbers, CT(local matrix), vary locally and significantly. The authors address this issue by using a mean for CT(mat) that is varied globally in accordance with the standard deviation of the CT-number distribution

of the matrix. This is the equivalent of extracting one realization of the possible fracture geometries within the framework of the method and the known uncertainties, then simply dilating (CT(mat)+standard deviation) and closing (CT(mat)-standard deviation) this geometry. To determine the variance in the simulation results as a result of local variation of the unknown CT(local matrix) would require a local analysis. A local analysis would result in a large number of possible fracture geometries that likely could vary substantially from those estimated by applying a variation in CT(mat) globally. If the objective of the investigation presented by the authors is to develop a methodology for estimating the flow path characteristics – which can conceivably range from a singular main flow path, to multi-stranded flow paths, to uniform flow – as a function of fracture geometry, which is coupled to stress history, then the local variation in aperture will be the determining factor. Bulk measurements - such as the permeability - can easily be fit by a planar plate model, therefore matching bulk measurements to simulated bulk measurements does not provide any information on the flow path geometry characterization. Likewise, if the uncertainty in the aperture variation on the local scale is large, the uncertainty of the flow path characterization will also be large, and thereby defeating the purpose of the proposed methodology. From what is presented here a rather simple conclusion may be reached, if CT(mat) varies locally and significantly then the estimated sub-voxel fracture apertures determined using the MA method may be far too uncertain to be of use in characterizing flow path geometry. Of course, this conclusion could be reached without the investigation presented by the authors. Unless the above outlined concerns are addressed the investigation presented by the authors amounts to a computational exercise that is likely disconnected from the system of interest by non-trivial uncertainty.

Page 6, lines 11-12: The authors should show that five averaged scans reduces noise in comparison to a single image, how much does it reduce noise? Is it significant?

Page 6, lines 17-18: How is the resampling performed? Are the CT number values of the resampled voxels equal to the imaged voxel? Also do you mean one voxel to 16

voxels?

Figure 4: The linear calibration used was matched to known apertures (by use of spacers) that are far larger than those inferred for flow simulation, casting doubt on the extrapolation of the calibration to the apertures inferred.

Page 7, lines 30-32: The authors state that apertures affect 2-3 voxels, but assume only a single voxel measurement is necessary to obtain the aperture. Isn't this information loss? Wouldn't this have an effect on the estimated fracture aperture? Do the 2-3 voxels reference resampled or original image voxels?

Page 7, line 34: The authors state that the MA method works well. Based on what evidence?

Page 8, lines 11-13: Yes, higher porosity should reveal lower CT numbers, if the composition does not affect the CT numbers.

Page 8 lines 10-15: I am confused by this paragraph, are the authors suggesting that the CT numbers of the matrix can be used to determine the pore sizes? Given the matrix has a permeability several orders of magnitude smaller than the fracture, the flow in the matrix can be ignored. Flow in the matrix is then discussed on pg 10-11: Are the authors suggesting that there is significant flow in the matrix? The permeability assigned to the matrix using the non-sequitur method outlined on page 8 are arbitrary, CT numbers do not provide any information beyond an estimate of porosity if one assumes CT(local matrix) is known.

Page 12 lines 31-32: The authors state that "significant material heterogeneities appear to be negligible for the sandstone," which does not follow from the discussion on the significant variation in permeability that the simulations predict with small global changes in CT(mat). This is the opposite of negligible. Page 13 lines 9-10, the authors state that the simulations are unable to reproduce the measured permeability hysteresis. This statement certainly follows from the results, and indicates major issues with

the practicality of the proposed methodology.

Page 12 line 13: The authors state, "one may assume stress-dependency of CT(mat)", why? The densities might change slightly due to compression, but the strain is likely to be very small, and the resulting effect on CT number is likely to be indiscernible.

Page 14-15 bulleted recommendations: None of these recommendations follow from the study outlined in the manuscript, instead they are generalized advice that may be useful, but is disconnected from the study. For instance, bulleted point 1, I agree pore-scale studies of heterogeneous porous rocks may be valuable, but this was in no way investigated in this manuscript. Again, bulleted point 2, I agree to better understand the effect matrix permeability has on fluid flow in fractured porous rock one may want to measure the matrix permeability before inducing a fracture, I don't think anyone would disagree. And, yes it would better to have higher resolution measurements of fracture roughness and aperture distribution.

After reading this manuscript it appears I would have been better off simply reading Huo and Benson, or the Watanabe papers, which are extensively referenced in lieu of analysis in this manuscript.

---

## Author Comment (AC1) · 20 May 2016

Dear reviewer,

We are very thankful for your efforts and constructive review. Please find attached a zip-file containing an elaborate response letter and a marked manuscript for a better allocation of possible changes resulting from your comments. The marked manuscript contains both, responses to you and the second reviewer. The according cross references you can find in your response letter.

The response letter addresses all your comments with the appropriate answers and suggests potential improvements in the manuscript.

[Figure]

We think that these changes to the manuscript will improve the quality and consistency of our paper and will meet the journal publication requirements.

With kind regards T. Kling (on behalf of the authors)

Please also note the supplement to this comment:
http://www.solid-earth-discuss.net/se-2016-41/se-2016-41-AC1-supplement.zip

---

## Author Comment (AC3) · 27 May 2016

Dear reviewer,

We found a further error in our manuscript which has to be mentioned and corrected.

Unfortunately, we interpreted the experimental results assuming an incorrect viscosity of the applied fluid so that determined permeabilities (between $10^{-12}$ and $10^{-13}$ m$^2$) do not represent actual permeabilities of the core sample which should be between $10^{-14}$ and $10^{-15}$ m$^2$. Correspondingly, we also assumed the same (wrong) viscosity as a boundary condition for our simulation so that simulated permeabilities also should be significantly lower.

We will revise the simulations and experimental results to fix that problem for the revised manuscript and will reconsider related parts of the discussion. First adjustments indicate that revisions indeed will change the absolute values of simulated and experimental results, however will barely affect their relative behavior so that it will not affect the central conclusions of the paper.

We deeply apologize for this mistake and try to fix it as fast as possible.

With kind regards T. Kling

---

## Author Comment (AC5) · 31 May 2016

Dear reviewer,

fluid flow simulations were re-simulated with adjusted fluid properties. As expected, new experimental and simulated results are lower than before, however do not affect their relative behavior and, hence, do not affect the general conclusions of the paper.

As we stated mistakenly, the experiments were not performed with water but with nitrogen. Indeed, nitrogen is not an incompressible fluid, however experimental conditions were chosen so that fluid flow simulations can be represented by a laminar, incompressible approach. The experiment was conducted at 2.1 MPa pore pressure so that

the slippage effect is highly suppressed and negligible. For analysis, common Darcy's law and a corrected one for gas showed identical results. Furthermore, repeated measurements with three different flow rates clearly showed Darcy flow behavior. There are several reasons for that such as the very small pressure drops due to the fracture, the low flow rates and the high pore pressure. So the compressibility in this experiment can be assumed as negligible and the simulation approach is also applicable for the actual fluid.

**We attached a new (preliminary) revised manuscript in the supplements. Following changes were performed:**

**Page 33, 35, 36, 37 and 38:** Adjustment of Figures 2, 5, 6, 7 and 8.

**Page 6, lines 4-9; Page 9, lines 17-19; Page 10, lines 9-10:** Adjusting information about experimental conditions and simulation input parameters for nitrogen.

**Page 16, lines 24ff:** Fracture permeabilities derived from cubic law approaches were corrected for matrix permeability to correctly reproduce the entire core permeability. Actually, the resulting range of permeabilities is significantly lower (cf. Figure 8) than mentioned in the initial revisions (Response 1). Hence, the cubic law approaches represent a valuable method to predict quantitative fluid flow, however are physically limited and cannot be used for qualitative flow analyses. This aspect is re-discussed in the new manuscript and is also adjusted in the summarizing sentence in the conclusion part (Page 19, lines 27-29).

The new simulation results do not affect the comparison with the comparative CT-based fluid flow simulations discussed on Page 15, lines 20ff or the error analysis.

We deeply apologize for the former mistake and think that the changes to the manuscript will improve the consistency of the manuscript and will meet the journal publication requirements.

With kind regards T. Kling

Please also note the supplement to this comment:
http://www.solid-earth-discuss.net/se-2016-41/se-2016-41-AC5-supplement.pdf
* * *
[Figure]

**Supplement:**

[revised manuscript text omitted]

Permeability measurements of the core sample are based on the steady-state method. After saturating the encapsulated sample with nitrogen ($N_2$), three different flow rates (12, 16 and 20 ml/min) are successively applied. A constant pore pressure of 2.1 MPa is applied to suppress the sleepage effect. Hence, the applied low flow rates and corresponding small pressure drops  reveal a linear relationship indicating laminar fluid flow. Hence, in this study the fluid can be treated like a incompressible fluid and permeability can be determined by using Darcy's law (Huo and Benson, 2015).

According to this, permeabilities are determined under stepwise changes of effective stress ($\sigma'$). Here, $\sigma'$ is defined as the difference between applied confining pressure and pore pressure. In order to characterize stress-dependency of the fractured sample comprehensively, changes in $\sigma'$ represent a full loading-unloading cycle. Following this, $\sigma'$ is increased stepwise from 0.7 MPa (2.1, 3.5, 5.5, 11.0 MPa) to 22.1 MPa under loading conditions and, subsequently, are decreased by applying equivalent stress intervals.

Contemporaneously, the core-holder is positioned in a medical X-ray CT scanner (General Electric Hi-Speed CT/I X-ray computed tomography) to reveal real-time images of the sample for every stress stage. Scans were performed at an energy level of 120 keV, a tube current of 200 mA and a display field of view of 25 cm. We use CT scans to obtain an x-y resolution in the plane of $0.5 \times 0.5$ mm² and a slice thickness of 1 mm resulting in corresponding voxel dimensions. As a result of the CT scans each voxel is assigned to a specific CT number in Hounsfield unit (HU). Furthermore, at each stress stage, multiple (five) scans are conducted and averaged afterwards, representing a practical method to reduce the random noise of CT scans by 50% as extensively discussed by Huo et al. (2016) and Pini et al. (2012).

**2.3 Image processing**

The geometry of the model is based on the averaged, multiple CT scan revealing a cylindrical set with a total dimension of $256 \times 256 \times 68$ voxels that still contains the multilayer construction of the core-holder as well as the filter plates of the fluid in- and outlet. Full processing is performed by using a customized MATLAB code. After reading the five data sets, every CT scan is resampled to an isotropic voxel size (one voxel of $0.5 \times 0.5 \times 1.0$ mm³ to 16 voxels of $0.25 \times 0.25 \times 0.25$ mm³) required for a proper computation of the CFD program. Subsequently, the five scans are averaged to a single image (Figure 3). In a further step, geometric information stored by single CT numbers of the (resampled) voxels are transformed voxel-wise to according geometric (local apertures) and hydraulic (local permeabilities) properties being essential for the flow simulations. The corresponding calibration approach is explained fully in the next paragraph. Finally, the core-holder and filter plates are numerically cropped based on known core dimensions and obvious density contrasts resulting in a final

sample dimension of 194 × 194 × 258 voxels. Note that marginal voxels of the sample are directly affected by the adjacent core-holder so that the processed sample is slightly smaller than expected (200 × 200 × 268 voxels).

Aperture calibration is based on the phenomena that the presence of (low-density) air or nitrogen, as used in this study, in  a homogeneous rock matrix reduces CT numbers of voxels containing present voids and also can affect adjacent voxels. Considering a fractured homogeneous rock along a cross section perpendicular to the fracture plane, the resulting density contrasts  can be perceived as a more or less pronounced anomaly depending on fracture width. Thus, an aperture calibration method (MA method) was developed by Johns et al. (1993) assuming that all X-ray attenuation is conserved in the CT image and that local apertures can be derived by integrating the available density anomalies. Dispersion of X-ray attenuation and partial volume effects can cause an expansion of the anomaly over adjacent voxels that gather this "missing attenuation" (Johns et al., 1993) and, in particular for larger fracture, represent a large portion of the entire anomaly. According to that, Johns et al. (1993) suggested a calibration-based linear relationship between aperture width and the integral of the full measured anomaly which was subsequently  confirmed in several fracture aperture studies (Bertels et al., 2001; Heriawan and Koike, 2015; Huo and Benson, 2015; Keller, 1997; Ketcham et al., 2010; Van Geet and Swennen, 2001; Vandersteen et al., 2003; Weerakone and Wong, 2010) and physically established by Huo et al. (2016). Accordingly, the linearity between missing attenuation ($CT_{MA}$) and fracture aperture $a$ can be simply described as:

$$CT_{MA} = C \cdot a \tag{1}$$

where the constant C is given by the slope of the calibration line. Furthermore, $CT_{MA}$ is defined as

$$CT_{MA} = \sum_{i=1}^{N}(CT_{mat} - CT_i) \tag{2}$$

where  $CT_i$ represents the CT number of the voxel along the cross section affected by the missing attenuation and N localizes the considered voxel. In this study, $CT_{mat}$ represents a idealized global threshold value for the matrix material assuming a homogeneous rock matrix as also assigned in previous studies (Johns et al. 1993; Keller, 1997; Keller et al., 1999). Indeed, using a global $CT_{mat}$ is a simplified assumption and provides additional errors particularly for heterogeneous rocks (Keller et al. 1997), however is sufficient for the intended straightforward purposes of this study. Usually, heterogenous rocks would require the usage of local $CT_{mat}$ values (e.g. Huo et al., 2016) which would result in a large number of possible solutions and would not improve the validity of the presented results. $CT_{mat}$ is determined by averaging the single modes of all CT numbers at every pressure stage, assuming that the most frequent CT number dominates the matrix of the rock sample. Our results reveal a relatively high matrix number of $CT_{mat} = 1862$ HU due to the high density of the sandstone, which is in line with other CT-based sandstone studies (Akin and

Kovscek, 2003; Huo et al., ; Vinegar et al., 1991).

Careful  calibration with different spacers (0.19, 0.29, 0.41, 0.52 mm) within the fracture indicates a slope of the linear calibration line of 5890 ± 38.3 HU/mm (Figure 4, Huo et al., 2016). According to Eq. (1) and (2) apertures, therefore, can be calculated by:

$$a = \frac{\sum_{i=1}^{N}(CT_{mat} - CT_i)}{5890 \pm 38.3 \ HU/mm} \tag{3}$$

As a consequence, Eq. (3) (with $N > 1$) can be used to describe local apertures along the fracture; however, it is not practical to model the entire core sample. Typically, the MA in voxels adjacent to the voxel containing the fracture depends on rock type and aperture size (Huo et al., (2016)). Considering several cross sections through the fracture used here (before resampling) indicates that occasional local apertures  affect 2-3 voxels, where the vast majority ($\geq 99\%$) of the attenuation is captured by the central voxel causing a local but marginal information loss. Hence, assuming that the main information about fracture aperture is stored in one voxel, we define $N = 1$, which also benefits the calculation of apertures for every single voxel of the core sample.  However, since the model is calibrated voxel-wise, the applied method will get more and more errorneous (which is also coupled to the CT resolution) with increasing fracture widths. This would cause artifactual apertures in the adjacent voxels actually representing the matrix and also would underestimate apertures in the voxels containing the fracture. In this study, most voxels containing the fracture range between 1700 and 1820 HU suggesting that calculated (and summarized) apertures of each aperture cannot exceed 0.15 mm and most widely are significantly smaller than 0.1 mm. This corresponds to Huo and Benson (2015), who determined mean apertures between 0.025 to 0.031 mm. Negative apertures due to $CT_i > CT_{mat}$ as a result of the heterogeneity of the matrix or due to image noise are defined as "zero apertures" with $a = 0$ mm.

Although, this strategy does not describe the basic intention of the common MA method, it provides a convenient solution to include data for the entire core and also enables integrations of  regions with higher porosity within  a homogenous matrix material where fluid inclusions should decrease $CT_{mat}$. Hence, voxels representing sections of the matrix with  significant porosity should reveal lower CT numbers ($CT_i < CT_{mat}$) so that these voxels, in simplified terms, are treated as equivalent apertures. These equivalent apertures represent the effective hydraulic diameter, which can be

[revised manuscript text omitted]

20   Furthermore, additionally enhanced fluid flow is observed in single parts within the matrix (Figure 6). Comparing with the CT image (Figure 3) indicates that this flow occurs along laminations containing darker matrix voxels with $CT_i < CT_{mat}$ according to Eq. (3). In homogeneous media, $CT_{mat}$ simply should represent the predominant mineral phase. In fact, the rather heterogeous sandstone in this study is dominated in quartz (detrital grains and cement), however is also enriched in feldspar. This feldspar component predominantly consits of plagioclase with minor alkali feldspars (Weissbrod and Sneh,

25   2002). Although, there are compositional differences, quartz and plagioclase typically reveal similar and, compared to most other minerals, relatively low $CT_i$ values (Ketcham, 2005; Tsuchiyama et al., 2005). Thus, the assumed $CT_{mat}$ (1862.6 HU) most widely can be ascribed to these dominant mineral phases. Hence, regions with significantly lower $CT_i$ values are caused by significant porous regions. This porosity heterogeneity is also observed in thin sections (Huo et al., 2016, Huo and Benson, 2015). However, the simulated overall matrix permeability can be assumed to be rather low which is also clarified

30   by carefully examining the propagation of the pressure fields corresponding to the predefined bulk matrix permeability of $10^{-19}$ m². Concurrently, propagation of the pressure fields also reinforces the assumption that major fluid flow occurs along the fracture. Furthermore, comparing absolute changes in fluid flow (Figure 6) due to loading between the lowest (0.7 MPa) and the highest (22.1 MPa) pressure stage indicates that most changes within the sample occur along the fracture plane while the simulated matrix flow remains nearly equal. Unfortunately, core permeability is derived by solving the Navier-Stokes-

[revised manuscript text omitted]

Furthermore, in order to be valid and as introduced in Sect. 2.3, the MSMA method requires a linearity of the calibration line and which also justifies the extrapolation of apertures as performed hereas reinforced by a careful calibration according to Huo et al. (2016) who also physically derived the linear relationship which justifies the extrapolation of apertures as performed here. On the other hand, Mazumder et al. (2006) described a nonlinear relationship between apertures and $CT_{MA}$

~~for fractured coal samples using an optimized MA approach. The derived calibration line indicates an increase of the slope with decreasing apertures, which would cause significant deviations in aperture calibration compared to a linear approach. However, available calibration data of our study are rather linear, so that simulations that are based on nonlinear regressions do not significantly differ from the linear approaches.~~

An additional factor that can contribute a major part to the simulated deviations is represented by $CT_{mat}$ describing an averaged threshold value for the matrix material. Being apparent from Figure 3, there are significant material heterogeneities within the sample. While darker voxels (with $CT_i < CT_{mat}$) can be assumed to contain more porous regions as discussed above, there are also significant brighter voxels (with $CT_i > CT_{mat}$). These significant higher $CT_i$ (partially $CT_i > 1900$ HU) values can be ascribed to the local accumulation of alkali feldspars, typically revealing much higher CT values (up 100 HU) than quartz due to the high attenuation of potassium (Ikeda et al., 2000). Thus, locally present alkali feldspar in a voxel can cause a local underestimation or even closing of the calibrated local aperture according to Eq. (3), most likely making a significant contribution to the underestimation of permeabilities a lower pressures (cf. Figure 7a).

[revised manuscript text omitted]

Additionally, Figure 8b represents a comparison of the simulation results with another economical approach based on the cubic law to predict fracture permeabilities. Fracture permeabilities are predicted empirically by applying various prominent approaches based on the averaged mechanical aperture ($a_m$) of distributed local apertures and their standard deviation (SD) for every single pressure stage (Amadei and Illangasekare, 1994; Barton and de Quadros, 1997; Lomize, 1951; Louis, 1967; Patir and Cheng, 1978; Renshaw, 1995). Fracture and entire core permeabilities, taking matrix permeability ($5.92 \times 10^{-19}$ m²) into account, are derived according to Huo and Benson (2015) who applied a more precise MA approach using local $CT_{mat}$ on the identical sample. As apparent from Figure 8b, the results of this alternative prediction approach reveal a wide range of possible permeabilities, which also approximate the experimental values. Similar to the simulations with a reduced global $CT_{mat}$ (with 1862.6 − 17.7 HU) permeabilities are significantly underestimated at low confining pressures but rather approximate actual permeabilities at higher pressures. Hence, the cubic law approaches also represent a valuable method to

predict quantitative fluid flow, however are physically limited by comprising flow channeling and related hydraulic properties such as connectivity or tortuosity and cannot be used for qualitative flow analyses.

Accordingly, the introduced MSMA method represents a further approximating step for successful CFD simulations, but also exposes current limitations. Although applying the method with a global $CT_{mat.}$ affirm a loss in accuracy as predicted by Keller (1997) for smooth fractures (< 35 μm), simulation results still are valuable approximate actual permeabilities in such fractures. Indeed, in this study compositional heterogeneities provide inaccuracies as predicted by Keller (1997), however are not that dominant causing a complete loss of information as indicated by the kinematics of several reproducible flow channels. Nevertheless, inaccuracy is not only provided by compositional heterogeneities, but also by sub-grid scale features particularly regarding higher confining pressure and, thus, closing local apertures.

[revised manuscript text omitted]

Despite the quantitative deviations, the simulated permeabilities indicate stress-dependency of the sample represented by a slight decrease in permeability with increasing effective stress and even imply hysteretic behavior. Besides minor calibration errors, the deviations are mainly caused by compositional heterogeneities of the matrix material and resolution-caused limitiations. The former facilitates underestimation of local hydraulic properties, while the latter prevents an accurate capturing of sub-grid scale features. Both error sources affect the reproduction of actual connectivity playing an important role in smooth fractures. Furthermore, the simulation is very sensitive to the choice of an adequate threshold value $CT_{mat}$ (1862.6 HU in this study). Small deviations from the ideal $CT_{mat}$ (±17.7 HU in this study) can cause enormous changes in simulated permeability by up to a factor of 2.6 ± 0.1. Thus, $CT_{mat}$ has to be defined with caution and can cause additional problems for rocks with significant mineralogical heterogeneities.

Nevertheless, A comparison with a  CT-based CFD-study by Watanabe et al. (2011) and various cubic law approaches, based on a rather recommended MA method using local $CT_{mat}$ values, reveals that the introduced MSMA approach can be valuable method to analyse quantitative and qualitative fracture flow. Considering the aforementioned comparative study a list of recommendations for future research is compiled, including a systematic investigation concerning different model and experimental setups, rock types, fracture modes as well as validation techniques.

~~simulated permeabilities, however enables the generation of a recommendation list for future research including: (a) Extensive, preliminary porosity studies and "blank tests" with the unfractured core to expose more detailed hydraulic properties of the sample. (b) "Blank tests" by applying a PEEK core-holder should reveal the influence of image noise when using a metal core-holder for the experiments. Additionally, high-resolution measurement techniques should indicate possible effects of small scale (<0.5 mm) fracture morphologies that are generalized due to the CT resolution. (c) The development of PEEK core-holders that are able to resist experiments under reservoir conditions. (d) A direct comparison of the applied MA and PH method to consider the merits and demerits of both approaches focussing on different fracture widths and types, different matrix materials and CT resolutions. (e) A more detailed analysis of the aperture calibration for the MA method by focussing smaller apertures (< 0.1 mm). (f) The development of new contrast agents or utilization of alternative technologies to validate qualitative fluid flow within the fracture.~~

**Data availability**

Since the size of the underlying data is too large for an upload, the authors encourage interested readers to contact the co-authors. Raw CT data can be obtained from DH (dhuosu@gmail.com). Processed CT data and simulation results (e.g. flow and pressure patterns) are stored on a server at the University of Mainz (Contact: enzmann@uni-mainz.de)

**Acknowledgement**

This study was mainly carried out within the framework of the Helmholtz Association of German Research Centres (HGF) portfolio project 'Geoenergy' and is part of the comprised reservoir engineering cluster. In addition, we acknowledge the postdoctoral grant to JOS, which was funded within the frame-work of DGMK (German Society for Petroleum and Coal Science and Technology) research project 718 "Mineral Vein Dynamics Modelling". The latter is funded by the companies ExxonMobil Production Deutschland GmbH, GDF SUEZ E&P Deutschland GmbH, DEA Deutsche Erdoel AG and Wintershall Holding GmbH, within the basic research program of the WEG Wirtschaftsverband Erdöl- und Erdgasgewinnung e.V. The authors also want to thank Rani Calvo from the Geological Survey of Israel for providing the Zenifim sandstone sample used in this study. In particular, we thank to Math2Market providing the GeoDict software package for fluid flow simulations. Furthermore, the authors are grateful to the Karlsruhe Institute of Technology (KIT), whose policy relating to open access journals facilitates financial support. Last but not least, we thank the anonymous referees for their constructive comments and valuable hints.

[revised manuscript text omitted]
. In (b) core permeabilities obtained by applying simple parallel plate models on more precise aperture data based on local CT$_{mat}$ values (Huo and Benson, 2015) are compared to simulated and experimental data.** **. ~~Literature data are based on a three granite core samples with a single, tensile fracture (SF), multiple, natural fractures (MF) and a single, tensile fracture measured by using an improved core-holder (SF PEEK). Furthermore, the effectiveness of the simulations is shown in (b) as factors, describing the discrepancy between each simulated and corresponding experimental permeability value, versus the normalized effective stresses depending on the highest stress stage applied during the associated experiment.~~**